# Impact of Temperature on the Bioactive Compound Content of Aqueous Extracts of *Humulus lupulus* L. with Different Alpha and Beta Acid Content: A New Potential Antifungal Alternative

Ulin A. Basilio-Cortes [1], Olivia Tzintzun-Camacho [1], Onecimo Grimaldo-Juárez [1], Dagoberto Durán-Hernández [1], Adabella Suarez-Vargas [2], Carlos Ceceña Durán [1], Alexis Salazar-Navarro [1], Luis A. González-Anguiano [1] and Daniel González-Mendoza [1,*]

[1] Instituto de Ciencias Agrícolas, Universidad Autónoma de Baja California, Carretera a Delta, Ejido Nuevo León s/n, Mexicali 21705, Baja California, Mexico

[2] Agro-Biotecnological Department, Universidad Tecnológica de Mineral de la Reforma, Camino Providencia—La Calera, No. 1000, Col. Paseos de Chavarría, Chavarria 42186, Hidalgo, Mexico

[*] Correspondence: danielg@uabc.edu.mx

**Abstract:** Hops contain a wide variety of polyphenolic compounds with diverse antimicrobial properties. This study aimed to evaluate the impact of temperature on the bioactive components of samples of aqueous extracts of hops with different characteristics. A central compound rotating design model was used in order to obtain optimal conditions of temperature and extract concentration to inhibit *Fusarium oxysporum* and *Alternaria solani*. At intermediate temperatures according to the design of experiments, significant effects on antifungal activity were observed. The optimal conditions with antifungal activity were at a concentration of 160 mg/mL and a temperature of 65 °C to obtain mycelial diameters $\leq$ 25 mm. The bioactive compounds were shown in the FT-IR spectrum after each heat treatment of both samples; significant changes were observed in the bands between 2786 to 3600 cm$^{-1}$ and 1022 to 1729 cm$^{-1}$. The content of total phenols and flavonoids showed a concentration increase of 4.54 to 6.24 mg GAE/g and 6.21 to 8.12 mg QE/g from an initial evaluation temperature of 25 °C to 57.5 °C, respectively, benefited by the heating temperature, enhancing antifungal activity. However, when increasing the temperature $\geq$90 °C, a tendency to decrease the concentration of bioactive compounds was observed, probably due to their denaturation due to the effect of temperature and exposure time, being non-thermolabile compounds at high temperatures. These aqueous extracts are an alternative to effective natural antifungals.

**Keywords:** *Humulus lupulus* L.; FT-IR; antifungal activity; aqueous extract; optimization

## 1. Introduction

Microorganisms of the genus of filamentous such as *Fusarium oxysporum*, *Rhi-zoctonia solani*, *Sclerotium rolfsii*, *Sclerotinia sclerotiorum*, and *Alternaria brassicicola* among others are phytopathogens worldwide, causing great damage and losses in agricultural production of different varieties of crop plants and even of postharvest fruits and vegetables with blight disease [1–4]. The main prevention used for decades has been the indiscriminate use of chemical fungicides (e.g., benomyl, carbendazim, thiabendazole, and alliete); however, they have caused pathogens to generate resistance in addition to promoting environmental contamination in crops, further potentiating the food insecurity and human health [5–7].

Effective natural compounds with activity to inhibit the development and growth of pathogenic microorganisms against fungi and bacteria are innovative trends for the immediate replacement of potentially toxic and environmentally harmful chemicals. Postharvest fruit conservation aims to extend shelf life and maintain a safe product. Various investigations have reported effective biomolecules from plant extracts due to their low cost and high performance, propitiating this effect due to the content of bioactive compounds and

essential oils that are the main precursors to presenting biomolecules with antimicrobial, antifungal, antioxidant, anticancer, and antihypertensive activity, among others [8–12]. The extraction of bioactive compounds by means of a previous thermal treatment has been studied to obtain better biological activities. Temperature-assisted extraction increases extraction efficiency over the cellular structure of the plant matrix, due to increased cell membrane permeability and breakdown of secondary metabolites from matrix interactions, thereby increasing transfer solubility mass of the bioactive compounds in the solvent. Increasing the temperature on the solvent could also decrease the surface tension and, consequently, improve the hydration of the plant material, resulting in a more efficient extraction. A high temperature decreases the viscosity of the extraction medium, promoting the penetration of the solvent into the plant particles and resulting in an improved and accelerated extraction process [13,14].

Natural products derived from plants are a highly viable option as biorational antifungal products, being environmentally friendly, high yield, and low cost, through restricted, reasonable, and rotating use with other natural compounds so as not to generate resistance against pathogenic microorganisms [15,16]. A viable option is the hop plant *H. lupulus* belonging to the *Cannabinaceae* family; its main use is to brew beer because it gives bitterness and characteristic aroma. This is due to the amount of α and β acids present in the variety of hops, which can have variations depending on climatic conditions, altitude, type of irrigation, luminosity, and soil nutrients, among other factors [17]. Hop acids α-(humulones) exhibit antifungal activity. Antimicrobial compounds including xanthohumol (XN), isoxanthohumol (IX), desmethylxanthohumol, α, β-dihydroxanthohumol, 6-prenylnaringenin (6-PN), and 8-prenylnaringenin (8-PN) have been recently isolated as broad-spectrum agents against microorganisms (e.g., bacteria, virus, fungi, and protozoa) [18–20]. In this context, the hop extracts could be a new antifungal agent for the control of diseases caused by *F. oxysporum* and *A. solani* species. The objective of this study was to obtain optimal conditions for the extraction of bioactive compounds from two varieties of hops with different alpha and beta acid content to potentiate the inhibition of the growth of phytopathogenic microorganisms.

## 2. Materials and Methods

### 2.1. Materials

In this study, two pellet hop samples of American and European origin of the varieties SummitMT and Huell Melon GermanyMT, Type 90 Hop Pellets with Lot P91-ZLUSUMO223 137 and Lot 17-518, respectively, harvest 2017 of the Yakima Hops® brand. Table 1 presents their characteristics. The reagents used were Meyer brand analytical grade.

**Table 1.** Compounds present in hop samples.

| Compounds | Hop Sample | |
|---|---|---|
| | **Summit$^{MT}$** | **Huell Melon Germany$^{MT}$** |
| α ácidos | 17.50% | 6.10% |
| β ácidos | 6.50% | 9.90% |
| Aceites totales | 3.0 mL/100 g | 1.1 mL/100 g |
| B-pinene | 0.80% | 0.30% |
| Myrcene | 45% | 20% |
| Linalool | 0.40% | 0.20% |
| Caryophyllene | 14% | 2% |
| Farnesene | 1% | 8% |
| Humulene | 22% | 1% |
| Geraniol | 0.60% | 0.20% |
| Others | 38.20% | 57.60% |

## 2.2. Microorganism

Two pathogenic fungi were isolated from cotton plants with root rot symptoms collected from cultivated fields in Mexicali Valley, Mexico. These isolates were identified using morphological characteristic and classified as *F. oxysporum* and *A. solani*, respectively.

## 2.3. Aqueous Extract of H. lupulus L.

The aqueous extracts of both varieties of hops were obtained using the methodology described by Nionelli et al. [21]. A total of 100 g of pellets were weighed, macerated independently in distilled water (1:1 *p/v*) until coated in a sealed amber container, and allowed to stand for three days. The extract obtained was filtered (Whatman No. 2 paper) and concentrated at room temperature until excess water was removed, recovering the extract obtained in a crystallizer for later analysis.

## 2.4. Aqueous Extract of H. lupulus L. at Different Temperatures

For both hop samples independently, an extract sample was placed in a 25 mL tube at a proportion of 1:10 *m/v*. It was later placed in a water bath according to the design of experiments (Table 2). Periodic mixtures were made with the help of a vortex every 5 min for 90 min in order to guarantee the temperature of the extract in the sample. Finally, the samples were applied in the antifungal activity (Table 3).

**Table 2.** Actual and coded values of experimental design.

| Independent Variables | Coded Values | | |
|---|---|---|---|
| | −1 | 0 | +1 |
| Temperature °C—$X_1$ | 25 | 50 | 90 |
| Concentration mg/mL—$X_2$ | 125 | 500 | 1000 |

**Table 3.** Central composite rotatable design with two factors: temperature (°C) and aqueous extract concentration (mg/mL) for both hop samples on in vitro antifungal activity against *F. oxysporum* and *A. solani*.

| Run | Extract Obtained at Different Temperature (°C) | Extract Concentration (mg/mL) |
|---|---|---|
| 1 | 41.25 | 562.5 |
| 2 | 25 | 1000 |
| 3 | 57.5 | 562.5 |
| 4 | 25 | 125 |
| 5 | 73.75 | 562.5 |
| 6 | 57.5 | 562.5 |
| 7 | 90 | 1000 |
| 8 | 90 | 125 |
| 9 | 57.5 | 781.25 |
| 10 | 57.5 | 343.75 |
| 11 | 57.5 | 562.5 |

## 2.5. Antifungal Activity of Aqueous Extracts In Vitro

The antifungal activity of the hop extracts against two phytopathogenic fungi, *F. oxysporum* and *A. solani*, was evaluated, following the methodology described by Arruda et al. [18] with some modifications. Culture media were prepared with standard methods agar (BD Bioxon® agar). The media were sterilized (121 °C, 0.1 MPa, 15 min) and poured into Petri dishes. After solidification, an aliquot of 1 ml of the extract was added independently according to the experiment design (Table 1) and spread over the surface until absorbed by the agar. The microorganisms were inoculated with a fragment of mycelium (3 mm in diameter); it was placed in the center of the plate to observe its growth and inhibition on the surface of the agar in the Petri dish. Then it was incubated aerobically at 25 °C for 7 days. Finally, on the seventh day, the diameter of the micellar fragment developed was

measured; boxes with agar without extract were used as a positive control. Three replicas were run simultaneously.

### 2.6. Determination of Total Phenols

Phenolic compounds were quantified by the spectrophotometric method using the Folin-Ciocalteu reagent according to Arruda et al. [18] with some modifications; 3.0 mL of Folin-Ciocalteu 1:10 (*v/v*) (Sigma-Aldrich, St. Louis, MO, USA). After 5 min of standing in the dark, 1.5 mL $Na_2CO_3$ solution (7.5 N) was added to stop the reaction. The absorbance was determined after 60 min against the blank (96% ethanol, *v/v*) at 760 nm in a spectrophotometer (DR6000$^{TM}$ UV VIS Spectrophotometer, Hach, IA, USA). The total phenol content was determined using a gallic acid calibration curve (0–200 mg/L) (Sigma-Aldrich, USA), and a linear correlation was obtained ($R^2$ = 0.9985). The results were expressed as gallic acid equivalents in milligrams of gallic acid equivalents per gram of dry sample (mg GAE/g).

### 2.7. Determination of Total Flavonoids

The total content of flavonoids was determined by spectrophotometry according to Arruda et al. [18] with modifications. From each aqueous extract, previously diluted in 96% ethanol (1:5 *v/v*), an aliquot of 0.5 mL was collected, and then 2 mL of distilled water and 0.15 mL of 5% $NaNO_2$ solution were added. After 5 min, 0.15 mL of 10% (*v/v*) $AlCl_3$ was added, and, 1 min later, 0.5 mL of 1 M NaOH was added to the mixture. Subsequently, 1 mL of distilled water was added and, with the help of a vortex, mixed for 30 s. The absorbance was measured at 510 nm in a spectrophotometer (DR6000TM UV VIS Spectrophotometer, USA). Total flavonoid content was determined using a standard catechin curve (0–200 mg/L) (Sigma-Aldrich, USA), and a linear correlation was obtained ($R^2$ = 0.9971). The blank was prepared as described above, without the addition of AlCl3. The results were expressed as catechin equivalents in milligrams of catechin equivalents per gram of dry sample (mg EQ/g).

### 2.8. Fourier Transform Infrared (FTIR)

A sample extract was placed on the ATR crystal surface. The FTIR spectra were obtained between 4000 and 500 cm$^{-1}$ using a Agilent 4300 Handheld FTIR (Santa Clara, CA, USA), with 40 scans and 4 cm$^{-1}$ resolutions in front of a background of the clean and empty ATR diamond crystal measured in the same instrumental conditions, three spectra of each sample being obtained. After each solution measurement, the ATR crystal was cleaned with ethanol and a soft cellulose paper, checking that the baseline was recovered before the introduction of another sample.

### 2.9. Statistical Analysis

All experiments were performed in triplicate. The results obtained were analyzed by one-way analysis of variance (ANOVA) using Origin 8.0 software (Origin Inc., Las Vegas, NV, USA). Mean differences were determined by Tukey's test ($p \leq 0.05$).

Optimization Experimental Design

To obtain the optimal conditions for the aqueous extract, a central compound rotating design model was used, composed of two factors: extract obtaining temperature (25, 38, 51, 64, 77, 90 °C) and extract concentrations (125, 300, 475, 650, 825, and 1000 mg/mL) for both hop samples (SummitMT and German Hüll MelonMT) as shown in Table 3.

The data was analyzed using the Response Surface Methodology (RSM) with De-sign-Expert 11 software version 7.1.5 and were fitted to polynomial regression model as shown in Equation (1):

$$Yi = b_0 + b_1X_1 + b_2X_2 + b_{12}X_1X_2 + b_{11}X_1^2 + b_{22}X_2^2 \tag{1}$$

where Y is the response variable, $X_1$ is the temperature (°C) at which the extracts are obtained, and $X_2$ is the concentration of the extract used to observe the in vitro antifungal activity against *F. oxysporum* and *A. solani*. The importance of the models was checked by analysis of variance (F test). Optimal conditions were obtained using the response surface method, where the selected response was hop variety (polyphenols) and in vitro antifungal activity.

## 3. Results

### 3.1. Effect of Temperature on the Antifungal Activity of Hop Extracts

The optimization of the antifungal was carried out to determine the best conditions of the aqueous extract (temperature °C and concentration mg/mL) evaluated with both hops and microorganism (Figure 1). To obtain the best conditions, the values of the independent variables were processed to quadratic models, since it is suggested to select the highest order polynomial where the additional terms are significant, and the model is not aliased. The factors coded from the equation were used to predict the response of each factor that is involved in this extraction process. According to the selected levels in a particular factor, high level factors were coded as +1 and low-level factors were coded as −1 (Table 2). The relative impact and the coefficient of the individual factor in the extraction process were determined using eq. 1. The result confirmed that the 95% confidence level showed precision and a good fit for the proposed model. The values of the $R^2$ coefficient of the technique were <0.05, which indicates that this model shows significant results. In addition, this value represented the agreement between the observed and predicted values of the quadratic model. The F value of this model was 10.15, and the proposed model was indicated to be significant. The F-value error was 0.01% and may have occurred due to noise. The *p* value was 0.05 and was less than the F value, indicating that the terms of the proposed model were significant.

The response variables were micellar diameter in vitro against *F. oxysporum* and *A. solani*. The lowest values (mycelial diameters) were used for both responses, since they showed a significant inhibitory effect ($p \leq 0.05$). Figure 1a–d shows the response surface of the antifungal activity of both aqueous extracts at different concentrations and temperatures: (a) SummitMT/*F. oxysporum* (FS) hops, (b) Hull Melon GermanyMT/*F. oxysporum* (FM) hops, (c) SummitMT/*A. solani* (AS) hops, (d) Hull Melon Germany MT/*A. solani* (AM) hops according to the experimental design (Table 2) for both hop samples with different α-acid content. The optimal conditions foreseen under the design of experiments to obtain significant antifungal activity with smaller mycelium diameters were the following: 65 °C temperature, a concentration of 160 mg/mL on *F. oxysporum* and *A. solani* with mycelium diameters predicted by the program for FS 18, FM 21, AS 14, and AM 16 mm. Experimentally, the optimal conditions were corroborated, obtaining diameters of mycelial for FS 22, FM 25, AS 18, and AM 19 mm. It should be noted that the diameters of the mycelial obtained as control were 62 and 56 mm for *F. oxysporum* and *A. solani*, respectively. On the other hand, in Figure 1a–d of the response surface graphs, a linear behavior was observed with respect to the concentration of the extracts as a function of temperature, presenting larger diameters of mycelia at a concentration of (125 mg/mL—25 °C), evidencing a lower inhibition on the microorganisms evaluated. FM showed sensitivity to high concentrations from temperatures of 38 to 77 °C (Figure 1b), presenting a significant effect on the concentration of extract as a function of temperature, evidencing an efficient antifungal activity; being that the hops from Hull Melon Germany MT contain a lower amount of α acids, this suggests that the compounds present in this hop are thermolabile, favoring the potentiation of inhibitory effects. While a similar behavior was observed in the FS sample, it did not present a significant effect (Figure 1a), since it exhibited antifungal activity from concentrations ≥300 mg/mL at temperatures from 38 to 77 °C. AS presented a higher antifungal activity compared to AM, showing efficient inhibition due to the effect of the concentration ≥650 mg/mL depending on the temperature (Figure 1c,d). In this case, the thermolabile compounds of the Summit

MT extract present a greater effect on *A. solani*; this proves that the different antifungal compounds from the hop extract will be effective depending on the microorganism to be inhibited.

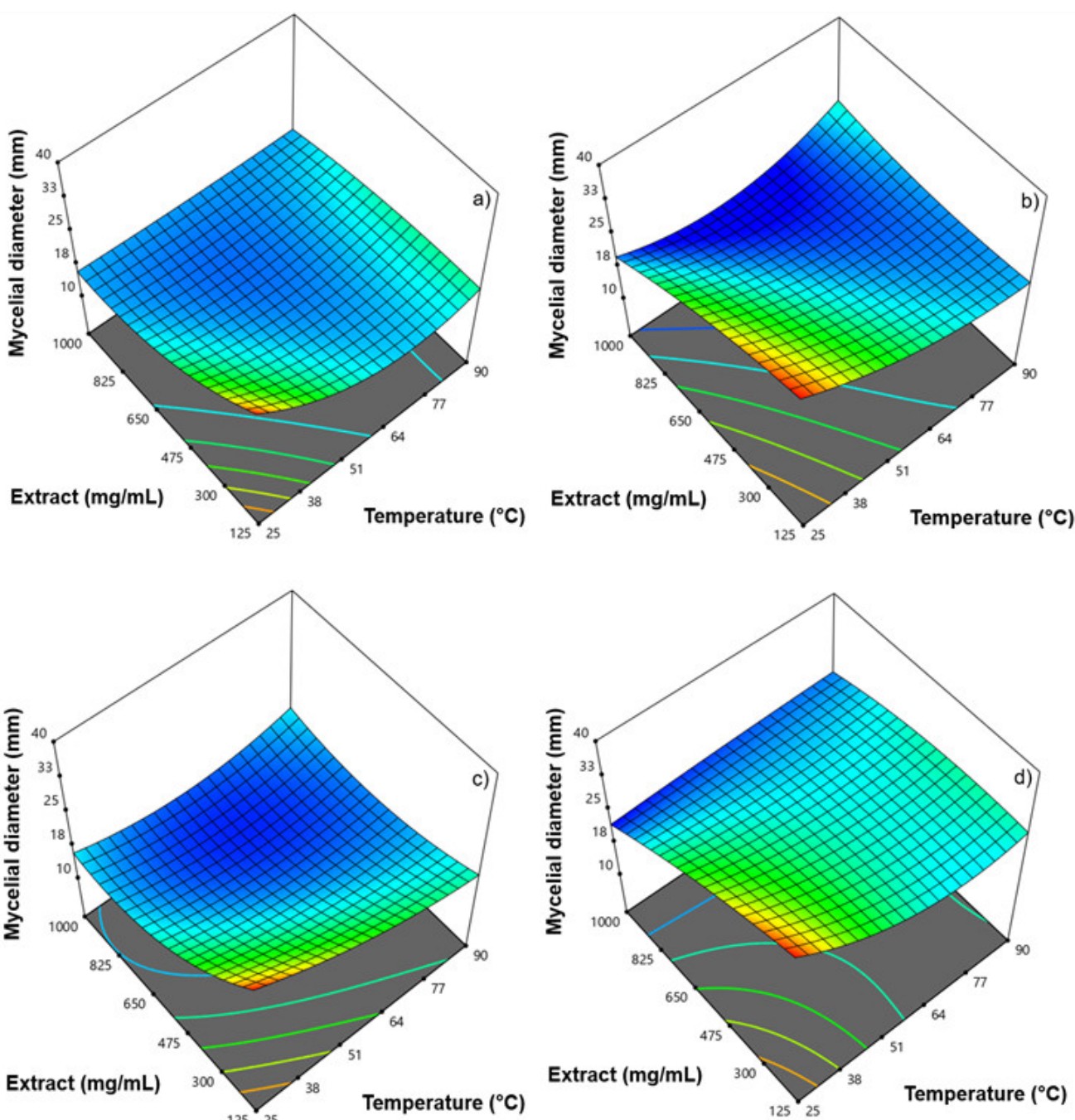

**Figure 1.** Antifungal activity of aqueous extracts at different concentrations and temperatures: (**a**) Summit[MT]/*F. oxysporum* (FS) hops, (**b**) Hull Melon Germany[MT]/*F. oxysporum* (FM) hops, (**c**) Summit[MT]/*A. solani* (AS) hops, (**d**) Hull Melon Germany[MT]/*A. solani* (AM). Mycelium diameter (mm) in relation to the concentration and temperature of the aqueous extract.

### 3.2. Effect of Temperature on Bioactive Compounds

Regarding polyphenols, four main chemical classes have been found in hops (e.g., flavan-3-ols, flavanols, phenolic carboxylic acids, gallic acid, vanillic acid, coumarin, flavonoids, tannin, and prenylflavonoids among others) [18,22] which exhibit efficient biological activities. However, this work shows that these compounds can potentiate their biological activity at a certain temperature depending on the variety of hops. In contrast, the aqueous extracts of both varieties of hops investigated (SummitMT and Hull Melon GermanyMT) showed an increase in the concentrations of phenols and flavonoids at a temperature of 57.3 °C, respectively (Table 4), with respect to the temperature of 25 °C, showing significant differences $p \leq 0.05$. The Hull Melon GermanyMT variety presented the highest values of flavonoids compared to the Summit hops, presenting a decrease in bioactive compound when receiving a thermal treatment of $\geq 57.5$ °C. These results show that the content of $\alpha$ acids has an effect on the heating temperature to obtain greater biological activity and potentiate an effect towards a productive benefit, as presented in this work with the antifungal activity evaluated in vitro, showing that the effect is directly proportional to the compounds of the thermally treated aqueous extracts.

**Table 4.** Total phenol content (TPC) and total flavonoid content (TFV) of aqueous extracts of two varieties of hops with different content of $\alpha$ acids.

| Temperature (°C) | Total Phenols (mg GAE/g) | | Total Flavonoids (mg QE/g) | |
|---|---|---|---|---|
| | **Summit**[MT] | **Hull Melon Germany**[MT] | **Summit**[MT] | **Hull Melon Germany**[MT] |
| 25 | 33.09 ± 0.34 [a] | 4.54 ± 0.74 [a] | 12.95 ± 0.12 [a] | 6.21 ± 0.12 [a] |
| 41.25 | 35.75 ± 0.21 [b] | 5.95 ± 0.29 [b] | 13.14 ± 0.12 [b] | 7.86 ± 0.09 [b] |
| 57.5 | 38.42 ± 0.71 [c] | 6.24 ± 0.41 [c] | 13.84 ± 0.12 [b] | 8.12 ± 0.18 [c] |
| 73.5 | 31.03 ± 0.09 [d] | 3.49 ± 0.84 [d] | 11.21 ± 0.12 [c] | 5.22 ± 0.14 [d] |
| 90 | 28.62 ± 0.47 [e] | 3.21 ± 0.35 [d] | 9.84 ± 0.12 [d] | 4.41 ± 0.31 [e] |

Means with different letters (a, b, c, d, e) within the columns differ significantly by Tukey's test ($p \leq 0.05$) between each sample and evaluation temperature.

### 3.3. Fourier Transform Infrared (FTIR)

Figure 2 shows the FTIR spectra of hop extracts from both samples that were heated to different temperatures according to the static design. The extracts presented an absorption band between 3000 and 3600 cm$^{-1}$ when increasing the heating temperature; a greater broad band was observed compared to the other treatments at lower heating temperatures, being that the optimal conditions of the extracts at a heating temperature of 65 °C presented a greater broad band that corresponds to the vibration of the OH groups attributed to the presence of phenols that formed hydrogen bonds or extraction solvents [23–25].

The spectrum shows two absorption bands between 2800 and 3000 cm$^{-1}$ that correspond to the C-H bonds of compounds with the presence of $CH_2$ and $CH_3$ groups [25]. In this work it was observed that neither band appears after heat treatment of the samples at $\geq 60$ °C. The transmittance band close to 1600 to 1300 cm$^{-1}$ corresponds to the estimated vibration of aromatic compounds (C-C), while the transmittance bands close to 1390 cm$^{-1}$ may be related to the presence of C-O groups of chlorophyll and phenolic compounds [25].

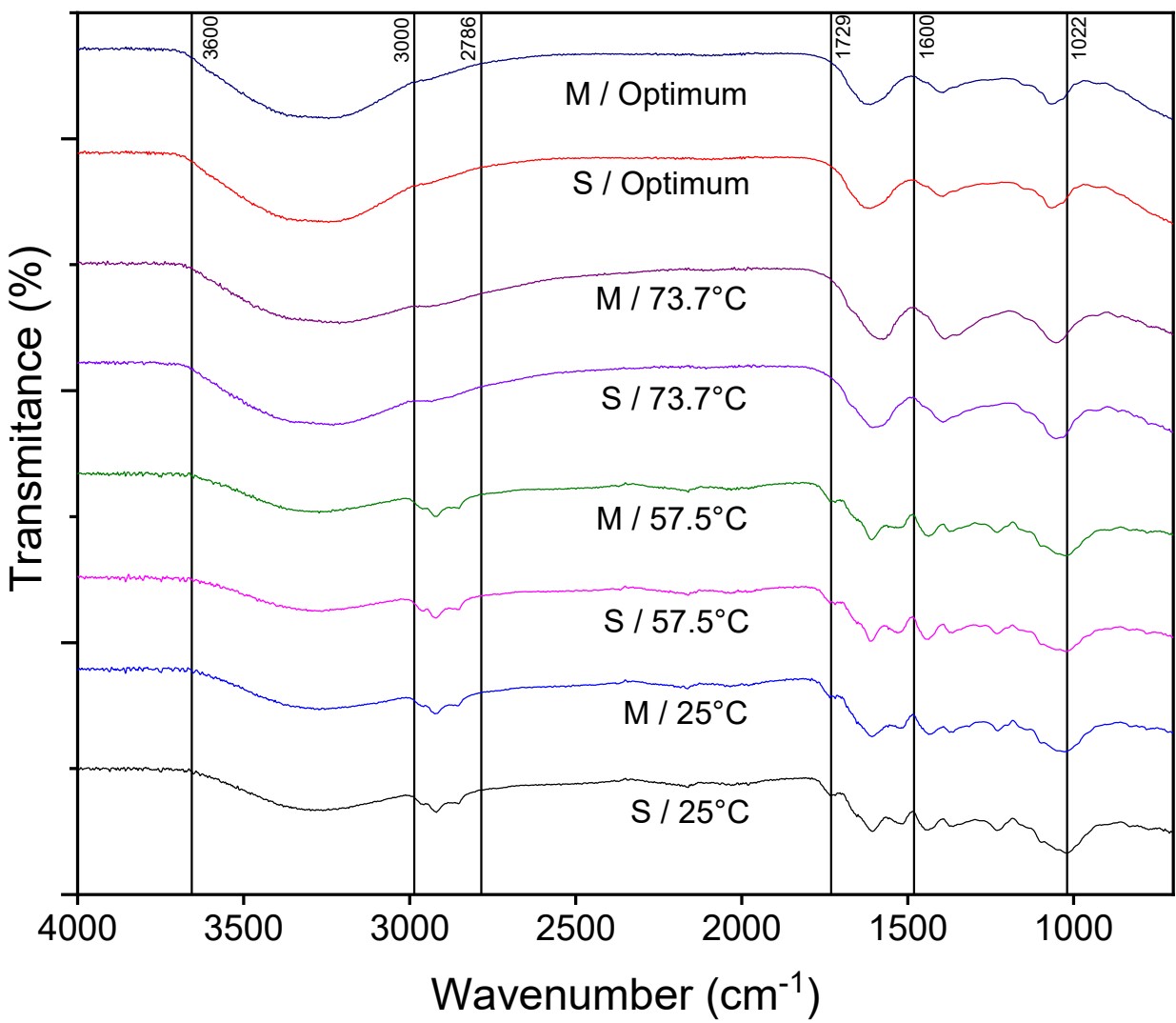

**Figure 2.** FTIR spectra of hop extract samples SummitMT (S) and Hull Melon GermanyMT (M).

### 4. Discussion

*F. oxysporum* and *A. solani* are worldwide phytopathogens causing great agricultural damage mainly in crops, vegetables, and fruits because they synthesize secondary metabolites to inhibit beneficial microbial consortia on plants, while in fruits and vegetables they create disruption on plant cells in order to guarantee their competence and pathogenicity [1,2,26,27]. Therefore, this research suggests that the volatile compounds generated by *F. oxysporum* are degraded and/or inhibited by bioactive α-acidic compounds (e.g., cohumulone, humulone, adhumulone) and β-acid (e.g., lupulones, colupulone, myrcene, caryophyllene, humulene, farnesene) that impart bitterness and the main volatiles in hops such as methyl octanoate, germacrene B, β-myrcene, geraniol, linalool, trans-α-bergamotene, α-cubebene, caryophyllene, cis-β-farnesene, α-humulene, β-selinene, and β-citronellol among others terpenoids [10,28]; these latter highly volatile compounds can be associated as inhibitory precursors for the development of both pathogens evaluated, by being thermally pretreated at low heating temperatures; however, it is known that the volatile bioactive compounds of hops kept at temperatures ≥ 90 °C for prolonged periods cause the loss of these volatile compounds, which is reflected in cases of brewing; the organoleptic profile is modified due to high boiling temperatures in the cooking of the beer, presenting the loss of volatile β-acids and the

production isomerization in the α-acids, promoting the enhancement of the bitter of the beer, being an indication to cause an effect on the antifungal activity [10,29]. Some bacteria such as *Bacillus subtilis*, *Streptomyces globisporus,* and *Paenibacillus polymyxa* produce volatile secondary metabolites (benzenes and ketones) with a fungicidal effect such as 6-methyl-2-heptanone, acetophenone, 2-pentylfuran, 2,5-dimethylpyrazine, benzothiazole, 5-methyl-2-hexone, and 2-heptanone to inhibit *F. oxysporum* and *A. solani* [30–33]; however, it has disadvantages when using these effective secondary metabolites, due to their high volatility, low yields, and stringent extraction conditions.

Hop cultivation factors such as the cultivation area, altitude, relative humidity, and phytopathological stress, among others, influence the production, accumulation, and preservation of secondary metabolites in the hop cones [18,34,35]. A study by Santarelli et al. [36] used high-power ultrasound extraction methods and high pressures to increase concentrations of bioactive compounds compared to the conventional method; they used a heating temperature of 60 °C, which presented an increase in bioactive compounds compared to the temperature of 25 °C. These results reinforce this work showing that bioactive compounds with the help of temperature and a certain exposure time promote molecular synergisms. Taking into account the previous evidence, the tendency to decrease the amount of bioactive compounds due to exposure to boiling temperatures for prolonged times is caused by oxidative reactions, possibly by the denaturation of bioactive molecules. This assertion is reinforced by works that show the effect of temperature on bioactive compounds from different botanical sources [37–39].

In the brewing industry, hops play an important role in brewing. It is common to find the term dry-hopping in articles and books, which consists of adding larger quantities of dry hops (pellet our flour) after the fermentation process has finished in order to reinforce the concentration of bioactive compounds that were degraded in the boiling process in addition to increasing aroma and bitterness, mainly for Indian Pale Ale styles in beer manufacturing, a promising technique for brewing functional beer from hop blends [10,40].

A study conducted by Bartmańska et al. [41] demonstrated that ethyl acetate, acetone, and methanol extracts of crude *H. lupulus* L. cones (hops) of the 'Magnum' variety collected in 2015 in the Lublin region (SE Poland) exhibited antifungal activity against *F. oxysporum*, *F. culmorum,* and *F. semitectum* using a concentration of 0.5 mg/mL, while the methylene chloride extract exerted antifungal activity against *Botrytis cinerea* at a concentration of 1 mg/mL. Nionelli et al. [21] used commercial Amarillo hop cones (Pinta, Disegna Group, Marostica, Italy), with a proximal composition of moisture, 11.0%; protein, 15.2%; fat, 3.4%; dietary fibers, 46.2%; total soluble carbohydrates, 2.0%; polyphenols and tannins, 4.7%; α-acids, 11.1%, and β-acids, 6.5%; obtaining an aqueous extract, which was boiled for 1 h in order to isomerize the hop acids, showing a significant inhibition of hyphal growth on: *Aspergillus parasiticus*, *Penicillium carneum*, *Penicillium polonicum*, *Penicillium paneum*, *Penicillium chermesinum*, *Aspergillus niger*, and *Penicillium roqueforti*; with temperature being an important factor to obtain inhibition, the isomerization yield of α-acids into iso-α-acids is invariably low and is also subject to variations; β-acids are even less soluble, and, therefore, their isomerization during the boiling of the wort is very low. The iso-α-acids resulting from the effect of temperature and hop variety are considered to be more abundant effective antimicrobial compounds [10]. Monoterpenes (e.g., myrcene, limonene, α-muurolene, σ-cadinene, β-pinene, limonene) present in hops have an antifungal effect against *Candida glabrata*, *C. albicans*, and *A. niger* [42].

The antimicrobial activity of hop extracts for the inhibition of bacteria is attributed to the action of the prenyl group of hop acids on the plasmatic membrane of the bacterial cell. In addition, it has been shown that the hydrogenated double bond in the prenyl group increases the activity antibacterial [42]. Yan et al. [43] demonstrated the mechanism of action of isoxanthohumol (*H. lupulus* L.) on the antifungal effect against *B. cinerea*, demonstrating that the antifungal activity is mainly related to metabolism; it affected the metabolic process

of carbohydrates, destroying the tricarboxylic acid cycle, and hindered the generation of ATP by inhibiting respiration. Studies on the antifungal activity of hop components are still limited; however, in this work we highlight and promote that the antifungal effect is directly related to the $\alpha$, $\beta$ acid, volatile, phenolic, and flavonoid compounds, which are differentiated by the characteristics that both hop samples present on antifungal activity, possibly due to the protonophoric action of the quantity and chemical structure of the acids present in each variety of hops, inhibiting micellar development. Recently, there is a growing interest in the investigation of the possible use of natural products, such as plant extracts, which can be minimally harmful for the control of fungal pathogens in different agro-industrial areas. The antifungal activity of plant extracts is well documented, and it is postulated that it is the effect of their main compounds or a synergistic effect of several compounds that form a blend. These findings suggest that various hop extracts may be effective agents for the control of economically important plant pathogens, such as *F. oxysporum* and *A. solani*.

FTIR spectral response demonstrated that, at 1088 and 1050 cm$^{-1}$, it is possible to observe two bands that refer to C-O for phenols, while the band at 870 cm$^{-1}$ refers to the vibrational frequency of the CH$_2$OH groups of carbohydrates [23]. In the samples heated to $\geq$60 °C, a significant decrease in the bands between 1000 and 1650 cm$^{-1}$ is observed. In a temperature range of 25 °C to 57.5 °C in the hop samples, a greater number of signals was observed between 3000 to 2786, as well as in 1729 to 1600 cm$^{-1}$, demonstrating that the temperature did not significantly affect the compounds present in both hop samples; however, at optimum temperature conditions of 65 °C and above 73.7 °C, a decrease in signals was observed over the same range described.

The number of these bands and their intensities depend on the compounds present in the sample and the replacement of aromatic rings. It can be observed that the intensities of the characteristic bands for the presence of phenolic compounds are different for the samples with different thermal treatment, which may be related to the higher content of bioactive compounds in the hop extracts in relation to the heating temperature. The position of this band depends on the presence and location of other functional groups, being that the presence of the broad band between 3100 and 3500 cm$^{-1}$ and the bands within the range of 1100–1300 cm$^{-1}$ may indicate that the carbonyl group is derived from carboxyl groups or may be present in phenyl compounds [23,25]. Compounds such as phenols, flavonoids, terpenes, (e.g., xanthohumol, isoxanthohumol, desmethylxanthohumol, $\alpha$, $\beta$-dihydroxanthohumol, and 6-prenylnaringenin, 8-prenylnaringenin) among others contain polyphenyl derivatives that have OH and carbonyl groups, with xanthohumol being a compound that can be an isomer. This effect is due to heat treatment to obtain isoxanthohumol [10].

In summary, this study revealed that the content of $\alpha$, $\beta$ acids, and volatile compounds that are potentially active upon receiving a thermal pretreatment at moderate temperatures to obtain a significant antifungal effect depends on the microorganism to be inhibited due to its mycelial structure and defense mechanisms. The various inhibitory behaviors due to the extraction temperature depend on the soluble thermolabile molecules contained in the hops that possibly promote synergisms; however, when the heating temperature is exceeded, antagonisms or denaturation of bioactive compounds are probably generated, presenting a reduction in the total content of phenols and flavonoids, and, therefore, affecting antifungal activity.

**Author Contributions:** Conceptualization, U.A.B.-C., A.S.-V., O.T.-C. and A.S.-N.; methodology, O.G.-J., D.D.-H. and C.C.D.; software, L.A.G.-A.; formal analysis, D.G.-M. and U.A.B.-C.; writing—original draft preparation, U.A.B.-C.; writing—review and editing, D.G.-M. All authors have read and agreed to the published version of the manuscript.

**Funding:** This research was funded by Universidad Autónoma de Baja California.

**Institutional Review Board Statement:** Not applicable.

**Informed Consent Statement:** Not applicable.

**Data Availability Statement:** Data sharing is not applicable for this article.

**Acknowledgments:** A special thanks to laboratory technician Elihu Raziel Morán-Niebla for his support in the use of the biotechnology laboratory of the Institute of Agricultural Sciences (ICA) of the Autonomous University of Baja California (UABC). In addition, to the bachelor's degree student of ICA-UABC, Misael Camacho-Trejo, for his support in the preparation and conditioning of extracts of hops.

**Conflicts of Interest:** The authors declare no conflict of interest.

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
