# Peer review of "Impact of Temperature on the Bioactive Compound Content of Aqueous Extracts of Humulus lupulus L. with Different Alpha and Beta Acid Content: A New Potential Antifungal Alternative"

_2036-7481, doi:10.3390/microbiolres14010017_

Round 1
Reviewer 1 Report
The present study entitled “Impact of Temperature on the Bioactive Compound Content of aqueous Extracts of Humulus Lupulus L. with Different Alpha and Beta Acid Content: A New Potential Antifungal Alternative” is interesting but organization of manuscript needs significant changes. The changes are as above.
· Lines 24-26: should mention peaks difference for better understanding
· Abstract seems a discussion instead of present values of results. Authors should mention some facts/values for a better understanding.
· Line 38-41: various investigations……… whereas authors cited only one study….
· Line 52-58: authors should split the introduction section into paragraph and then arranged as: first paragraph should reveal the importance, need and background of study,
· Second paragraph should focus on the literature review of study, third with techniques used and last one with novelty, scope and objectives.
· Authors must mention the role of different temperature over the extraction of valuable compounds for antimicrobial potential. This is lacking in introduction section.
· Lines 69:71: objectives of the study should be written in more details.
· Lines 104-106: please write more details about methodology
· Line 107: Table 1 is not experimental design. Please check it carefully. Also experimental design of RSM is missing. In addition, this experimental design should have one heading of “experimental design”
· Lines 137-138: details of FTIR spectra are missing. How did authors prepared samples. Background removal etc.???
· Figure 2 needs more improvements; Y-axis has mixed axis. Please make it more wider to make it clear. In addition, peak indication is wrong: upper right corner, it should be 1500 cm-1 instead of 3000.
· From RSM, only phase has been completed while predicted values or optimum values generated by software has not been performed. Comment please. Regression coefficient and model values e.g., p values are missing.
Author Response
Reviewer #1:
The present study entitled “Impact of Temperature on the Bioactive Compound Content of aqueous Extracts of Humulus Lupulus L. with Different Alpha and Beta Acid Content: A New Potential Antifungal Alternative” is interesting but organization of manuscript needs significant changes. The changes are as above.
- Lines 24-26: should mention peaks difference for better understanding
Thank you for your valuable comments and suggestions.
Response: In response to the suggestion. the difference between the peaks in the significant bands of the FTIR analysis is described. See lines 24-26.
- Abstract seems a discussion instead of present values of results. Authors should mention some facts/values for a better understanding.
Thank you for your valuable comments and suggestions.
Response: Important values of the results obtained in the work were added. See line 22-28.
- Line 38-41: various investigations……… whereas authors cited only one study….
Thank you for your valuable comments and suggestions.
Response: One more quote (Review) is added in order to reinforce the information. See line 52, 62.
- Line 52-58: authors should split the introduction section into paragraph and then arranged as: first paragraph should reveal the importance, need and background of study,
- Second paragraph should focus on the literature review of study, third with techniques used and last one with novelty, scope and objectives.
Thank you for your valuable comments and suggestions.
Response: The paragraphs were organized for a better understanding and coherence of the research work. See lines 36-79.
- Authors must mention the role of different temperature over the extraction of valuable compounds for antimicrobial potential. This is lacking in introduction section.
Thank you for your valuable comments and suggestions.
Response: A paragraph referring to the extraction of bioactive compounds due to the effect of temperature was added. See line 54-62.
- Lines 69:71: objectives of the study should be written in more details.
Thank you for your valuable comments and suggestions.
Response: A paragraph referring to the extraction of bioactive compounds due to the effect of temperature was added. See line 75-79.
- Lines 104-106: please write more details about methodology
Thank you for your valuable comments and suggestions.
Response: The methodology is described in detail.
- Line 107: Table 1 is not experimental design. Please check it carefully. Also experimental design of RSM is missing. In addition, this experimental design should have one heading of “experimental design”
Thank you for your valuable comments and suggestions.
Response: Corrections done and the table is added. See line 105. Table 3 represents the number of runs that the experiment design produced. See line 106.
- Lines 137-138: details of FTIR spectra are missing. How did authors prepared samples. Background removal etc.???
Thank you for your valuable comments and suggestions.
Response: Corrections done. See line 146-152
- Figure 2 needs more improvements; Y-axis has mixed axis. Please make it more wider to make it clear. In addition, peak indication is wrong: upper right corner, it should be 1500 cm-1 instead of 3000.
Thank you for your valuable comments and suggestions.
Response: Corrections done. See figure 2.
- From RSM, only phase has been completed while predicted values or optimum values generated by software has not been performed. Comment please. Regression coefficient and model values e.g., p values are missing.
Thank you for your valuable comments and suggestions.
Response: Once the optimal conditions were obtained, the experiments were carried out to corroborate the prediction of the program. See line 205-2011.
The values of regression coefficient, model and description was added to the manuscript. See line 177-191.

Reviewer 2 Report
Dear authors,
Thank you for a well written manuscript. A few minor comments that I feel will improve the manuscript:
Editing and formatting
Humulus lupulus – should be in italics.
52-53 – all scientific names should be in italics.
55-56 – …of early blight in tomato, potato, tobacco, among others…this sentence does not make sense – rewrite to make sense.
65 – all scientific names should be in italics.
ml vs mL – please be consistent.
160 – The optimization of the antifungal activity was carried out to determine
284-289 – sentence is too long with full stops where they do not belong. Re-write for better reading.
293 – F. culmorum and F. semitectum used for the first time here, not defined anywhere else.
Statistical analysis
Before performing an ANOVA with Tukey post hoc analysis was normality of data checked?
Results
In addition to figure 1, the inhibition diameters obtained for antifungal activity should be added for better understanding of the data.
Conclusion
Can be improved upon.
Author Response
Reviewer #2:
Dear authors,
Thank you for a well written manuscript. A few minor comments that I feel will improve the manuscript:
Editing and formatting
Humulus lupulus – should be in italics.
Thank you for your valuable comments and suggestions.
Response: Corrections done.
52-53 – all scientific names should be in italics.
Thank you for your valuable comments and suggestions.
Response: Corrections done.
55-56 – …of early blight in tomato, potato, tobacco, among others…this sentence does not make sense – rewrite to make sense.
Thank you for your valuable comments and suggestions.
Response: Corrections done. See line 37-40.
65 – all scientific names should be in italics.
Thank you for your valuable comments and suggestions.
Response: Corrections done.
ml vs mL – please be consistent.
Thank you for your valuable comments and suggestions.
Response: Corrections done.
160 – The optimization of the antifungal activity was carried out to determine
Thank you for your valuable comments and suggestions.
Response: Corrections done. The values of polyphenols and antifungal activity were used to obtain optimal conditions. See line. 170-172.
284-289 – sentence is too long with full stops where they do not belong. Re-write for better reading.
Thank you for your valuable comments and suggestions.
Response: Corrections done. Paragraphs are separated for better understanding. See line. 170-172.
293 – F. culmorum and F. semitectum used for the first time here, not defined anywhere else.
Thank you for your valuable comments and suggestions.
Response: Corrections done. 299 and 304.
Statistical analysis
Before performing an ANOVA with Tukey post hoc analysis was normality of data checked?
Thank you for your valuable comments and suggestions.
Response: The data was verified, as well as the statistical design data for the optimization.
Results
In addition to figure 1, the inhibition diameters obtained for antifungal activity should be added for better understanding of the data.
Thank you for your valuable comments and suggestions.
Response: The inhibition values are in figure 1 in a range of 10 to 40 mm
Conclusion
Can be improved upon.
Thank you for your valuable comments and suggestions.

Reviewer 3 Report
“Impact of Temperature on the Bioactive Compound Content of Aqueous Extracts of Humulus Lupulus L. with Different Alpha and Beta Acid Content: A New Potential Antifungal Alternative” is the title of the proposed article. The manuscript describes the studies aimed to evaluate the impact of temperature on the bioactive components of samples of aqueous extracts of hops with different characteristics. Authors describe antifungal activity, TPC and TFV assay and FTIR spectra of aqueous extracts at different concentrations and temperatures.
The research has been carried out properly, the manuscript is clearly written and the results are of interest. Some little revision as follow and the manuscript can be published.
Title
Line 3: "Lupulus" must be written in lowercase.
Abstract
General consideration: it should be better explained immediately that reference is made to the extraction temperature. May authors improve the sentence of the abstract?
Introduction
General consideration: the objective of the study should be written a little better.
Line 43: please, “Humulus lupulus” should be written in italic.
Lines 52-53: Please, use italic format for all fungi names.
Line 65: a) All fungal names should written in italics. b) authors should write Fusarium oxysporum as "short" binomial name e.g., F. oxysporum.
Line 71: “In vitro” should be written in italic format.
Material and Methods
General consideration: authors should standardize hop names. For example, starting from line 75 if MT mean trademark, “Huell MelonTM (Germany)” and “SummitTM” can be the hop names in the article.
Line 82: please, write fungi names as "short" binomial name e.g., F. oxysporum and A. solani.
Line 84: please, write hop name as "short" binomial name e.g., H. lupulus L.
Line 91: please, see line 84.
Table 1: authors should check (a) language, (b) greek symbols and (c) the percentage of components (the sum is a little bit more than 100). How do explain this? Please insert more informations.
Line 102-103: please, see line 82.
Line 153: please, see line 71.
Line 154: please, see line 82.
Results
General consideration: please, use the standardized hop names (see line 75).
Line 162: most probably "phytopathogenic fungi" is better than generic "microorganisms".
Line 165-166: please, see line 82.
Line 167: authors probably should write mycelium instead micelle, please correct it.
Line 168: micellar is wrong. Did authors want to write mycelial diameter? Please, correct it.
Line 212: please, see line 71.
Table 3: please, check bold formats and hop names.
Figure 2: please improve quality of the figure, especially the Y-axis.
Section 3.3: please, check the wavenumber (cm-1) and formulas’ subscript.
Discussion
Line 268: please, use short binomial name e.g., A. solani.
Line 291: please, use short binomial name.
Line 293: please, use short binomial name e.g., F. oxysporum.
Line 293: F. culmorum and F. semitectum are first time cited, use full binomial name e.g., Fusarium culmorum and Fusarium semitectum.
Line 313: please, use short binomial name e.g., H. lupulus L.
Line 328: Please, use short binomial name.
Author Response
Reviewer #3:
“Impact of Temperature on the Bioactive Compound Content of Aqueous Extracts of Humulus Lupulus L. with Different Alpha and Beta Acid Content: A New Potential Antifungal Alternative” is the title of the proposed article. The manuscript describes the studies aimed to evaluate the impact of temperature on the bioactive components of samples of aqueous extracts of hops with different characteristics. Authors describe antifungal activity, TPC and TFV assay and FTIR spectra of aqueous extracts at different concentrations and temperatures.
The research has been carried out properly, the manuscript is clearly written and the results are of interest. Some little revision as follow and the manuscript can be published.
Title
Line 3: "Lupulus" must be written in lowercase.
Thank you for your valuable comments and suggestions.
Response: Corrections done.
Abstract
General consideration: it should be better explained immediately that reference is made to the extraction temperature. May authors improve the sentence of the abstract?
Thank you for your valuable comments and suggestions.
Response: Corrections done.
Introduction
General consideration: the objective of the study should be written a little better.
Line 43: please, “Humulus lupulus” should be written in italic.
Thank you for your valuable comments and suggestions.
Response: Corrections done
Lines 52-53: Please, use italic format for all fungi names.
Thank you for your valuable comments and suggestions.
Response: Corrections done
Line 65: a) All fungal names should written in italics. b) authors should write Fusarium oxysporum as "short" binomial name e.g., F. oxysporum.
Thank you for your valuable comments and suggestions.
Response: Corrections done
Line 71: “In vitro” should be written in italic format.
Thank you for your valuable comments and suggestions.
Response: Corrections done
Material and Methods
General consideration: authors should standardize hop names. For example, starting from line 75 if MT mean trademark, “Huell MelonTM (Germany)” and “SummitTM” can be the hop names in the article.
Thank you for your valuable comments and suggestions.
Response: The names were left with the MT prefix, because there are varieties with similar names but which are harvested in other parts of the world and obviously contain a different amount of bioactive compounds. In order to replicate this work for future research, it is necessary to correctly identify the raw material.
Line 82: please, write fungi names as "short" binomial name e.g., F. oxysporum and A. solani.
Thank you for your valuable comments and suggestions.
Response: Corrections done
Line 84: please, write hop name as "short" binomial name e.g., H. lupulus L.
Thank you for your valuable comments and suggestions.
Response: Corrections done
Line 91: please, see line 84.
Thank you for your valuable comments and suggestions.
Response: Corrections done
Table 1: authors should check (a) language, (b) greek symbols and (c) the percentage of components (the sum is a little bit more than 100). How do explain this? Please insert more informations.
Thank you for your valuable comments and suggestions.
Response: Names and language were revised, being the correct way to describe themselves.
The percentages are directly proportional with respect to the dilution ratio for each ounce, these values were obtained directly from the batch on the official website of the raw material supplier: https://www.yakimachief.com/commercial/hop-varieties.html
Line 102-103: please, see line 82.
Thank you for your valuable comments and suggestions.
Line 153: please, see line 71.
Thank you for your valuable comments and suggestions.
Line 154: please, see line 82.
Thank you for your valuable comments and suggestions.
Results
General consideration: please, use the standardized hop names (see line 75).
Thank you for your valuable comments and suggestions.
Line 162: most probably "phytopathogenic fungi" is better than generic "microorganisms".
Thank you for your valuable comments and suggestions.
Response: Corrections done. See line 110.
Line 165-166: please, see line 82.
Thank you for your valuable comments and suggestions.
Line 167: authors probably should write mycelium instead micelle, please correct it.
Thank you for your valuable comments and suggestions.
Response: Corrections done
Line 168: micellar is wrong. Did authors want to write mycelial diameter? Please, correct it.
Thank you for your valuable comments and suggestions.
Response: Corrections done
Line 212: please, see line 71.
Thank you for your valuable comments and suggestions.
Table 3: please, check bold formats and hop names.
Thank you for your valuable comments and suggestions.
Figure 2: please improve quality of the figure, especially the Y-axis.
Thank you for your valuable comments and suggestions.
Response: Corrections done
Section 3.3: please, check the wavenumber (cm-1) and formulas’ subscript.
Thank you for your valuable comments and suggestions.
Discussion
Line 268: please, use short binomial name e.g., A. solani.
Thank you for your valuable comments and suggestions.
Response: Corrections done
Line 291: please, use short binomial name.
Thank you for your valuable comments and suggestions.
Response: Corrections done
Line 293: please, use short binomial name e.g., F. oxysporum.
Thank you for your valuable comments and suggestions.
Response: Corrections done
Line 293: F. culmorum and F. semitectum are first time cited, use full binomial name e.g., Fusarium culmorum and Fusarium semitectum.
Thank you for your valuable comments and suggestions.
Response: Corrections done
Line 313: please, use short binomial name e.g., H. lupulus L.
Thank you for your valuable comments and suggestions.
Response: Corrections done
Line 328: Please, use short binomial name.
Thank you for your valuable comments and suggestions.
Response: Corrections done

Round 2
Reviewer 1 Report
no further comments from my side